# Neuro-Symbolic Inverse Constrained Reinforcement Learning

**Oliver Deane**    OLIVER.DEANE@BRISTOL.AC.UK  and  **Oliver Ray**    CSXOR@BRISTOL.AC.UK
*University of Bristol, Bristol, United Kingdom*

Editors: Leilani H. Gilpin, Eleonora Giunchiglia, Pascal Hitzler, and Emile van Krieken

## Abstract

Inverse Constrained Reinforcement Learning (ICRL) is an established field of policy learning that augments reward-driven exploratory optimisation with example-driven constraint inference aimed at exploiting limited observations of expert behaviour. This paper proposes a generalisation of ICRL that employs weighted constraints to better support lifelong learning and to handle domains with potentially conflicting social norms. We introduce a Neuro-Symbolic ICRL approach (NSICRL) with two key components: a symbolic system based on Inductive Logic Programming (ILP) that infers first-order constraints which are human-interpretable and generalise across environment configurations; and a neural system based on Deep Q learning (DQL) that efficiently learns near-optimal policies subject to those constraints. By weighting the high-level ILP constraints (based on the order in which they are learnt) and encoding them as low-level state-action penalties in the DQL reward function, we effectively allow earlier constraints to be overridden by later ones. Unlike prior work in ICRL, our approach is able to continue working when exposed to newly encountered expert behaviours that reveal more nuanced exceptions to previously learnt constraints. We evaluate NSICRL in a simulated traffic domain, which shows how it outperforms existing methods in terms of efficiency and accuracy when learning hard constraints; and which also shows the utility of learning defeasible norms in an ICRL context. To the best of our knowledge, this is the first approach that places equal emphasis on exploratory and imitative learning while also being able to infer defeasible norms in an interpretable way that scales to non-trivial examples.

## 1. Introduction

This paper presents a neuro-symbolic (NS) approach to Inverse Constrained Reinforcement Learning (ICRL). Our method generalises prior work in the field of Neuro-Symbolic Reinforcement Learning (NSRL) (Acharya et al., 2023) which has successfully integrated the interpretability of symbolic methods with the noise tolerance and scalability of neural networks in an RL setting. We extend those ideas with ICRL mechanisms that further enable the discovery of implicit constraints by combining exploration with knowledge extracted from expert demonstrations. This allows an agent to learn operational constraints within an environment without requiring exhaustive mathematical specification in advance (Scobee and Sastry, 2019; Liu et al., 2024).

Our proposed approach, Neuro-Symbolic Inverse Constrained Reinforcement Learning (NSICRL), combines NS methods with ICRL to learn a neural policy for efficiently operating in complex environments while simultaneously inferring constraints as high-level, human-interpretable symbolic rules. Representing constraints in a high-level symbolic language also allows for transferability across environmental configurations. For example, a constraint

like "Do not drive through a red light" can be reused in a variety of problem settings with varying road positions and junction layouts. Moreover, associating constraints with finite weights also enables us to represent defeasible 'norms' that facilitates lifelong learning by allowing newly acquired examples of expert behaviour to serve as exceptions to previously learned norms (without having to retrain the entire system from scratch). For example, after initially learning a norm "Do not drive off the road" from a limited set of expert observations, NSICRL can subsequently learn what could be considered an exception in the form of a more highly weighted norm "Drive off the road to avoid an obstacle" when later presented with an enlarged set of expert observations (as could easily happen in a lifelong or active learning setting or in domains with conflicting social norms).

In the remainder of this paper, we introduce relevant background concepts (Section 2), before outlining a formal implementation of the NSICRL system (Section 3). We validate efficacy with respect to prior research using a Traffic Simulator environment, including a demonstration of the utility of defeasible norm generation (Section 4). Finally, we discuss the key contributions of NSICRL in the context of related work (Sections 5) and discuss avenues for future research (Section 6).

## 2. Background

### 2.1. Reinforcement Learning

Reinforcement Learning (RL) provides a robust method for learning optimal policies through exploration that maximise cumulative reward over time Kaelbling et al. (1996). It is highly effective for solving complex, multi-step problems in dynamic environments and enables systems to make decisions that consider both immediate and long-term benefits (Mnih et al., 2013). RL problems are typically framed as episodic Markov Decision Processes (MDPs), where the goal is to learn a policy $\pi$ that selects the optimal action $a$ in a given state $s$ to maximise the expected discounted cumulative reward. Among various RL algorithms, a widely used approach is Q-learning, a model-free, value-based method that learns the optimal action-value function $Q^*$ and commonly follows a policy that selects actions based on the highest estimated value, i.e., $a = \arg\max_{a \in A} Q(s, a)$. The concept of Deep Q-Learning (DQL) extends Q-learning by using deep neural networks to approximate $Q^*$ in high-dimensional state spaces (Mnih et al., 2013). Traditional Q-learning relies on a tabular approach, which becomes infeasible as the state space grows. In contrast, DQL employs a Deep Q-Network (DQN) that approximates the Q-value function to map states to Q-values for each possible action.

### 2.2. Inverse Constrained Reinforcement Learning

To enforce compliance with environmental constraints, the MDP can be modified to a Constrained Markov Decision Process (CMDP) $M_C$. This extends a standard MDP by imposing a set of constraints $C \subseteq S \times A$, which restrict the set of valid actions in each state, where the valid action set in state $s$ is defined as: $A_C(s) = A(s) \setminus \{a \in A(s) \mid (s, a) \in C\}$ (Altman, 1999). Inverse Constrained Reinforcement Learning (ICRL) is a framework for learning these constraints from expert trajectories. For instance, Maximum Entropy Inverse Constrained RL (Scobee and Sastry, 2019) is a widely used method for discrete state-

action spaces; it first learns a policy in an unconstrained MDP, then iteratively infers constraints by identifying state-action pairs that are likely under the learned policy but have low probability in expert trajectories. Existing approaches represent constraints as individual state-action pairs, cost functions, or trajectory-based restrictions (Malik et al., 2021; McPherson et al., 2021; Subramanian et al., 2024). This hampers interpretability, particularly when managing a large set of constraints, and makes it difficult to discern which aspects of the expert observations influenced learning. Furthermore, as these representations are tied to specific state-action pairs or trajectory structures, they often fail to generalise to new configurations of an environment (Baert et al., 2023). Logic-Constrained Q-learning (LCQL) (Baert et al., 2023) is a extension of ICRL that begins to address this problem by using logic-based machine learning to learn symbolic constraints which were used to remove state-action entries from Q-table of a conventional RL system. But the use of an explicit Q-table significantly hinders scalability (to the extent LCQL cannot be practically applied to the examples in this paper) and their restriction to hard constraints prevents the incremental refinement of constraints and does not accommodate exceptions to previously learned norms. We are proposing to continue in this vein by using NS methods in order to overcome both these limitations of LCQL.

## 2.3. Inductive Logic Programming

Like (Baert et al., 2023), we use a logic-based machine learning approach called Inductive Logic Programming (ILP) to derive symbolic norms using the optimised policy and expert examples (Muggleton, 1991; Cropper et al., 2020). ILP infers a logic program (hypothesis) that explains observed data through formal rules and relations. It takes as input a background knowledge base, positive examples (instances the hypothesis should cover), and negative examples (instances it should exclude). ILP searches for a hypothesis that maximises coverage of positive examples while minimising coverage of negatives. The search space is restricted by a user-defined language bias, which limits the form and complexity of potential hypotheses. Here, we use the ILP system ALEPH, which employs a top-down search strategy with a "bottom clause" to constrain the search space (Srinivasan, 2001). We learn logical rules that define conditions for norm violating behaviour, thus we treat expert demonstration trajectories as negative examples, and policy-generated state-action pairs that deviate from expert behaviour (so-called seed constraints) as positives (see Section 3).

## 2.4. The Traffic Domain

We evaluate NSICRL using the SUMO Traffic environment in which agents must follow accepted norms and road rules, namely "stay on the road" and "do not enter junctions on a red light" (see Figure 1) (Lopez et al., 2018). This setting, used in previous research (Baert et al., 2023), extends the standard discrete gridworld benchmark for evaluating ICRL methods (Liu et al., 2022) by incorporating high-level concepts (roads and junctions). Expert examples are generated using the SUMO simulator which uses car control algorithms to generate realistic car journey trajectories. These trajectories are discretised to align with a grid-based set up. The resulting trajectories consist of a list of successive state-action pairs. A state consists of four values: the $(x, y)$ position of the agent, a traffic light signal

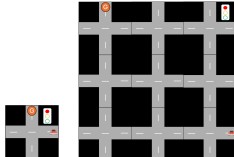

Figure 1: The original small (left) and our larger complex (right) Traffic environments.

value (0 or 1), and a binary indicator of whether the position lies on the road. Each action corresponds to movement in one of four cardinal directions (0-3) or stay still (4).

The equivalent logic representation of any given scenario comprises a set of logical predicates: $at/3$ denotes agent position, $tls/2$ represents the traffic light signal, and $go/2$ gives the agent's selected action. NSICRL also takes a Background Knowledge (BK) containing logical facts required for the ILP to induce hypotheses. For the Traffic domain, the $onRoad(X, Y)$ auxiliary predicate informs that a position $(X, Y)$ is on a road. The $atJunction(X, Y, D)$ predicates state that position $(X, Y)$ is immediately before a junction if travelling in direction $D$. The $move(X, Y, D, X1, Y1)$ predicate defines agent movements over a single time step where $X$ and $Y$ are original positions, and $X1$ and $Y1$ are subsequent positions after moving in direction $D$. These predicates are simple encodings of the environmental state along with the state transition rules represented in the DQL. To facilitate comparison with prior work, we use the state-space encoding of Baert et al. (2023), where $tls$ represents a global traffic light value where 1 means "Green" for cars on latitudinal roads (East-West) and "Red" for cars on longitudinal roads (North-South). Thus, to improve readability of hypotheses, we added a predicate, $tls\_local$, to define a normalised traffic signal - where a value of 1 means "Green" for any agent, irrespective of which road they are on (see Appendix B for the complete BK).

## 3. Implementation

In this section, we outline how the above concepts interact to form our NSICRL system, using the Traffic Domain as an example use case. Figure 2 depicts the overall structure; it takes as input a set of expert Trajectories $T$, a nominal (unconstrained) $MDP$, a background knowledge $B$ and a hypothesis space defined by a language bias $M$. The output is a set of logical constraints $C$ containing learned norms in the form of logical hypotheses $(H)$ induced at each iteration, as well as a constrained policy $\pi$.

NSICRL leverages two sub-modules to iteratively build and refine $C$. The first consists of a Deep Q Learner (DQL) which learns an approximate optimal policy for the given MDP[1]. Using a constraint inference method inspired by the Maximum Entropy ICRL (ME-ICRL) approach proposed by Scobee and Sastry (2019), NSICRL identifies state action pairs which are optimal according to this learned policy but are unlikely with respect to the set of expert trajectories. We consider these pairs 'Seed Constraints'. We use the ILP algorithm ALEPH as the induction module; using $T$ as negative examples and the seed constraint as positive, it generalises the seed to a logical hypothesis $H$ which captures the necessary conditions for a norm violation. All state-action pairs satisfying $H$ are subsequently used to update the

---

1. For small environments, this can feasibly be a single-layer approximate Q-learner (e.g., a perceptron).

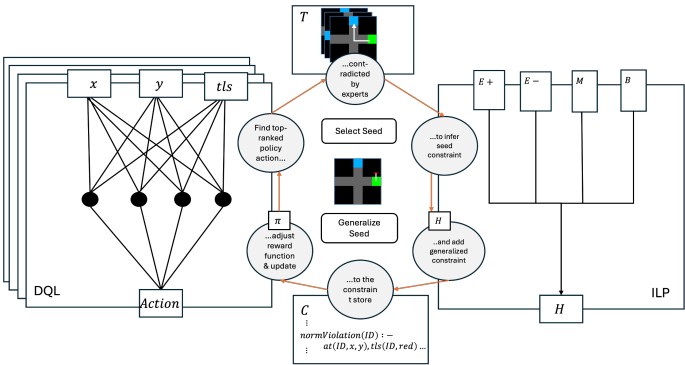

Figure 2: NSICRL interleaves a policy optimiser (DQL) and a constraint generator (ILP) to iteratively build a set of (defeasible) constraints (C). Seed constraints (visualised in the centre) are selected from the policy and generalised by ILP.

agent's nominal policy via an interaction with the policy optimiser's reward function. A new seed constraint is inferred from the updated policy, and this process continues as the set of logical constraints $C$ is iteratively augmented.

### 3.1. Inferring Seed Constraints

The DQL first learns an approximate optimal policy from goal-directed exploration using a nominal reward function. In the Traffic case, it receives state inputs as floats consisting of the agent's position, traffic light value, and a binary flag indicating whether the agent is located on a road. It outputs actions as discrete numerical encodings mapping to the four cardinal directions (0-3) or stay still (4): for instance, an agent selecting north when at position $(2,1)$ when the light is red is $[(2,1,0,0),1]$. The nominal reward function is derived from the simulated sumo environment whereby reward is delivered when the agent reaches a final goal state. To infer the Seed Constraint, we generate the set of all unique state-action pairs $T'$ from the set of expert trajectories $T$ whereby $T' = \bigcup_{\tau \in T}$. We order $T'$ by state visitation frequency to generate $T^{\mathrm{rank}}$, thus prioritising common constraints. Seeds are selected based on divergence from expert behaviour; for each state in $T^{\mathrm{rank}}$, the policy is queried for an optimal action, and if the action is not in $T'$, the associated state-action pair becomes the seed constraint. Grounding this in the Traffic example, a nominal policy would go directly to the goal, going off-road in the process; the action that takes the agent off road would be captured as a seed constraint because it is not contained in $T'$.

### 3.2. Hypothesis Induction

The hypothesis induction module, ALEPH, generalises seeds to a logical hypothesis $H$ which captures conditions for a norm violation. ALEPH takes a set of positive examples $E^+$, negative examples $E^-$, a logical program representing environment background knowledge $B$, and a search space $M$. $B$ is a set of logical facts defining relations and entities within the environment (for Traffic, this is $onRaod/2$, $atJunction/3$, $move/4$, see Section 2). Because we want to learn conditions for violations, $E^+$ consists of a seed constraint

represented by an $id$, where the state and action for that $id$ is encoded into the background knowledge and is logically represented by $normViolation(id)$. $E^-$ is the set of atoms of the form $normViolation(id)$ for every $id$ representing a state-action pair in the set of expert trajectories $T'$. $M$ and $E^-$ are initialised at the outset and are not updated.

Unlike the DQL which represents state-action pairs with floats, ALEPH requires input as first-order logic facts. We therefore define a mapping that translates between the two. For instance, $[(2, 1, 0), 1]$ would translate to logical form: $at(id, 2, 1).\ tls(id, red).\ go(id, north).$ ALEPH generalises seeds to a logical hypothesis $H$. $H$ comprises clauses taking the form $normViolation(ID)$ :- $b_1, \ldots, b_m$. The head is the target predicate applied to a variable (here, an ID representing a given state-action pair), and the body is a conjunction of literals defining the conditions required for the head to hold true. The ID represents a sample which consists a state and an action, as well as the other auxiliary information encoded in the background knowledge.

A representation of $H$ is used to update the DQL policy for use in future iterations. To achieve this, Selective Linear Definite (SLD) clause resolution derives all state-action pairs consistent with $B \cup H$, and the resulting pairs are mapped back to state-action space (hereby referenced as $SA_c$). $SA_c$ is subsequently used to update the DQL weights to ensure its policy abides by the novel rules in $H$.

### 3.3. Updating the Nominal Policy

We update the policy used for seed identification via an interaction with the DQL's reward function. During training, each constrained state-action pair satisfying $H$ is associated with a penalty term. This ensures the DQL's weights are updated to generate an approximate policy that avoids norm-violating actions in associated states. We include a constrained meta-policy which converges on a set of coefficients that mediate the relative strength of penalties. This applies a graded penalty to each $(s, a) \in SA_c$. The method learns a set of penalty values for state-action pairs associated with each $H$ (within the set of hypotheses $C$) that maximise the reward function while minimising violations, with the additional meta-constraint that penalties for state-actions associated with hypotheses induced in early iterations are smaller than those learned in recent iterations. As a result, NSICRL learns "defeasible" norms that can be overridden when updates to $T$ introduce exceptions and contradictions.

## 4. Results

### 4.1. Learned Hypothesis

When tested on the Traffic Simulation environment, the NSICRL loop delivers a hypothesis containing a high-level symbolic representation of the inferred norms:

$$\text{normViolation}(ID) \text{ :- } at(ID, X, Y), go(ID, D), move(X, Y, D, X2, Y2), \tag{1}$$
$$not(onRoad(X2, Y2)).$$
$$\text{normViolation}(ID) \text{ :- } at(ID, X, Y), go(ID, D), move(X, Y, D, X2, Y2), \tag{2}$$
$$atJunction(ID, X2, Y2)), tls\_local(ID, red).$$

Here, $ID$ is a variable standing for the identifier used to look up a given state-action within the background knowledge. Clause 1 discourages actions within state-actions that take the agent to an off-road state, irrespective of the agent's cardinal direction ($D$). Clause 2 prohibits the agent from moving into a central junction when the light is red ($tls\_local(ID, red)$).

## 4.2. Learning Hard Constraints

To validate NSICRL against prior work, we conduct experiments measuring norm violation rates and cumulative reward. We compare these results to similar methods for constraint inference in discrete domains: standard ICRL with Maximum-Entropy ICRL (ME-ICRL) and Logic-constrained Q-learning (LCQL) (Scobee and Sastry, 2019; Baert et al., 2023)[2]. First, we explore whether NSICRL can achieve equal performance for hard constraints in the small environments used in prior work (see Figure 1 (left)). We test optimised policies constrained by each method's inferred constraints, counting the number of times the agent following this policy violated a ground truth norm ("stay on road" and "don't cross junctions on red light"). For each method, we train 4 policies each reaching one of four goal states positioned at the north-, east-, south-, and west-most points on the road. We measure violation count after every iteration over 100 test runs. For NSICRL and LCQL, an iteration consists of inferring a new seed constraint, generalising it to a hypothesis and applying it to the policy (via graded penalties for NSICRL, and Q-table deletions for LCQL). For ME-ICRL, an iteration is the inference of a single state-action (seed) constraint applied to the policy with uniform penalties [3].

As illustrated in Figure 3a, NSICRL, similar to LCQL, successfully acquires all norms after an average of six iterations. Thereafter, their respective policies make minimal violations. As ME-ICRL learns at the state-action level, more iterations are required to achieve the same level of adherence. Similarly, NSICRL compares favorably when tested on its ability to transfer learned rules to new environments. We take the updated policy after each iteration and apply it to an alternative environment configuration where road and junction positions are altered. Figure 3b gives total constraint violations of each method when tested the on the novel configuration (over 100 runs). NSICRL matches LCQL in being able to transfer learned norm rules to the new environment (shown by immediate adherence), while ME-ICRL must re-learn from scratch. This transfer requires minimal manual intervention, needing only an update to the road and junction positions in the background knowledge.

Additionally, we demonstrate how NSICRL uniquely handles large state-action spaces while remaining interpretable (see Figure 1 (right) for a large complex grid). As shown in Figure 3c, for simple 5x5 grids, both methods effectively infer rule sets in reasonable time. However, as grid size increases beyond 25x25, LCQL becomes intractable due to the explosion of the state-action table.

---

2. We instantiate NSICRL with a perceptron as the DQL module. Experiments used a 2GHz Quad-Core Intel i5 CPU. Policies were trained in Python and executed on MacOS 15.3.1.

3. For all cases, we choose to only use a single seed constraint at each iteration.

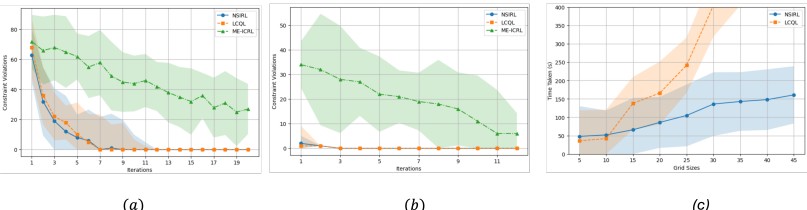

$(a)$ $(b)$ $(c)$

Figure 3: Left: Number of constraint violations with increasing iterations. Middle: Transfer Learning whereby NSICRL and LCQL have few violations in new configurations from outset. Right: Comparing NSICRL and LCQL in time taken to induce a final hypothesis for increasing grid sizes (we cut off after 15 minutes).

## 4.3. Learning Defeasible Constraints

A major advantage of our approach is that learned norms can be overridden by novel ones introduced by updates to the set of expert examples. We demonstrate this attribute using an extension of the traffic test domain. Given an initial set of expert trajectories $T$, an initial hypothesis is induced (see Clause 1). All trajectories within $T$ cover a small 5x5 grid. We then introduce an augmented set of expert trajectories $T+$ which represents a larger 10x10 grid. Crucially, the northern section of this enlarged environment contains an obstacle on the road; here, we envision this as a pothole that the agent should aim to avoid. To navigate this, a learning agent would have to violate the $stay - on - road$ constraint, leave the road, bypass the obstacle, and return to continue its journey to the goal.

Any method exclusively inducing hard constraints would not learn this exception. By using a graded penalisation factor applied to the reward function, NSICRL can learn the added constraint that stops the agent from entering the Pothole state:

$$\text{normViolation}(ID) \text{ :- } \text{at}(ID, X, Y), \text{before\_pothole}(X, Y, D), \text{go}(ID, D) \tag{3}$$

The action of leaving the road remains permissible, all-be-it with a penalty. Without requiring retraining from scratch, the system successfully learns to violate the past constraint as this is penalised less heavily than the newly-learned pothole constraint, and the learner can still continue on to learn an approximate optimal policy for reaching its goal.

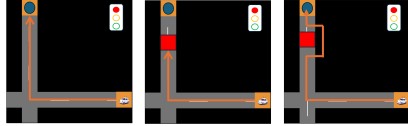

Figure 4: Pothole Scenario. Left: an agent reaches the goal (circle) following the initial 'Stay-on-road' constraint. Middle: Hard constraints inhibit navigation of the obstacle (red square). Right: Defeasible constraints allow for a policy that overrides previous constraint to proceed to the goal.

We demonstrate this experimentally using this pothole example, where ground truth norms are: "stay on road" and "do not enter potholes". The "stay on road" is defeasible meaning agents must never enter potholes and must only go off road when encountering potholes. As described, policies are first learned in a 5x5 grid before a 10x10 grid containing a pothole cell in the northern region is introduced. Policies are tested in the 10x10 grid and points are rewarded for reaching any of the 4 goals (north, east, south, and west).

A study into constraint violations revealed that NSICRL's policies achieve the desired behaviour at test time, only violating the "stay-on-road" norm when necessary and never entering a pothole state. An alternative method whereby penalties are enforced uniformly (e.g., ME-ICRL), does not handle this differentiation and violates each constraint with a probability of approximately 0.5. Figure 5 presents the cumulative reward at test time after each iteration (i.e., after each time an $H$ is induced) summed over all policies. NSICRL is able to navigate past the pothole to reach the northern goal (as well as all others). LCQL's hard constraints inhibit this navigation, meaning it does not achieve the same average reward because it never reaches the northern goal.

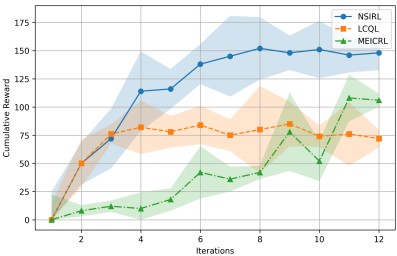

Figure 5: Defeasibility: NSICRL's policies navigate to all goals and maximises reward, while LCQL's hard constraints block access to the northern goal, limiting cumulative reward. As in Figure 3, MEICRL will require considerably more iterations to reach the same level of norm compliance.

## 5. Related Work

In the field of ICRL, many methods have leveraged the principles of Inverse RL to learn constrained policies operable in various environment settings (deterministic and stochastic) with different state-action representations (discrete and continuous) (e.g., Scobee and Sastry (2019); Malik et al. (2021); McPherson et al. (2021)). Typically, constraints are derived as a point in feature space which does not occur in expert data but would elicit greater performance. In essence, we build on this idea with inferred constraints that generalise over larger areas of feature space. A subset of ICRL research infers soft constraints by integrating a penalisation term and permitting occasional violations until cumulative penalties exceed a threshold (Gaurav et al., 2022; Papadimitriou et al., 2022; Subramanian et al., 2024). However, this is built for stochastic environments operating with aleatoric uncertainty and is not designed for overriding legacy constraints based on newly provided expert examples (as is proposed here). Further, existing methods do not focus on human-level interpretability;

an important differentiation as transparency is frequently considered a necessity in AI ethics and safety (e.g., Dignum (2017); Tubella et al. (2019)).

Existing ILP systems can derive general interpretable rules from observed trajectories, such as in Inductive General Game Play which infers rules from game traces (Cropper et al., 2020). However, it follows the closed-world assumption, treating unobserved atoms as false. NSICRL avoids this by using seed constraints as the positive set and not automatically invalidating state-action pairs outside the expert example set. Logic-Constrained Q-Learning is closely related to our work in that it uses logical inference to generalise constraints to high-level interpretable concepts (Baert et al., 2023). By enforcing constraints via a state-action table, they do not focus on incremental evolution of constraints, are restricted to learning hard constraints in small environments, and do not attempt to learn weighted constraints that can be overridden by newly-learned norms (see Appendix A for a complete method comparison table).

## 6. Discussion and Conclusion

In sum, we present a neuro-symbolic framework for learning social norms in the form of high-level, interpretable, symbolic constraints by combining autonomous exploration with expert imitation, in which exceptions to previously-learned constraints are enabled through an interaction with the agent's reward function. We demonstrate experimentally how NSICRL is able to a) match prior work in learning interpretable hard constraints that generalise across environment configurations, b) better scale to more complex state-action spaces, and c) handle conflicting norms with defeasible constraints.

We argue that symbolic representations of norms are essential for interpretability, editability, and transferability across environments. ILP is particularly well-suited for this compared to alternative propositional rule learners, as rules can be refined by users and can be enriched with background knowledge. This forms the foundation for future work to further leverage ILP's strengths — such as utilising recent interactive mechanisms proposed by Ray and Moyle (2021) that would enable users to intervene to shape hypotheses and prevent potentially lengthy chains of exceptions. Meanwhile, neural policy learners are essential for the exploration-driven component of NSICRL, as they have been demonstrably efficient and accurate in approximating complex value functions (Li, 2017). Notably, our evaluation was limited to a discrete, deterministic domain. Future work could extend NSICRL to continuous and relational domains to better leverage neural networks, exploiting research in logical neural networks to constrain the learned policy (Zambaldi et al., 2018; Riegel et al., 2020; Hoernle et al., 2022; Dang-Nhu, 2020).

Finally, we motivate our work as learning exceptions due to limited expert data. As highlighted by Neufeld et al. (2021), exceptions can also arise in the face of contradicting ethical principals where a system must handle such dilemmas by selecting a "lesser of 2 evils" option (Neufeld et al., 2021). For example, they provide a variation of Pacman in which an agent must learn not to eat scared ghosts, but should choose to do so when cornered by other forbidden states. Their method handles this by enforcing *pre-defined* defeasible logical programs. While such cases are beyond the scope of this paper, preliminary experiments suggest NSICRL is capable of learning the necessary programs, and by extension policies, to address them.

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

## Appendix A. Method Comparison Table

Table 1: Table comparing NSICRL to alternative ICRL methods

| Method | Interpretable | General | Continual Learning | Defeasible | Complex Envs |
|--------|:---:|:---:|:---:|:---:|:---:|
| ME-ICRL | | | | | ✓* |
| LCQL | ✓ | ✓ | ✓ | | |
| BICRL | ✓ | | | ✓ | |
| **NSICRL** | ✓ | ✓ | ✓ | ✓ | ✓ |

## Appendix B. Background Knowledge for the Traffic Environment

```
tls_local(S,V1) :- at(S,X,Y), atJunction(X,Y,D),
    (D=north;D=south), tls(S,V), V1 is V.
tls_local(S,V1) :- at(S,X,Y), atJunction(X,Y,D),
    (D=east;D=west),tls(S,V), V1 is 1-V.

move(X,Y,east,X1,Y) :- X1 is X+1.
move(X,Y,north,X,Y1) :- Y1 is Y+1.
move(X,Y,west,X1,Y) :- X1 is X-1.
move(X,Y,south,X,Y1) :- Y1 is Y-1.
move(X,Y,zero,X,Y).
atJunction(X,Y,D) :- move(X,Y,D,3,3).
onRoad(X,3).
onRoad(3,Y).
onRoad(X,7).
```

Listing 1: Full Background Knowledge for the Traffic Environment

