# OpenReview forum: "Neuro-Symbolic Inverse Constrained Reinforcement Learning"
_nesyconf.org/NeSy/2025/Conference — NeSy 2025 Poster_

### Official Review · Reviewer_epQC · 2025-03-23
**The paper proposes a novel method to learn and refine symbolic constraints from experience and expert demos, but lacks some experimental evaluation**

**Rating:** 5
**Confidence:** 5

**Review:**

The paper is well written and proposes an apparently original methodology to iteratively learn and refine symbolic constraints for RL agents. The method outperforms state of the art approaches which either remove constraint-violating state-action pairs, or do not discern among different constraints.
Overall, the methodology is interesting, but I think some more experimental results are needed:
1. The authors should compare to the more classical setting for constrained RL, where properly solves a constrained optimization problem is solved [1]
2. Cumulative reward should be reported as mean and standard deviation. Moreover, why isn’t ME-ICRL reported in Figure 6b?
3. What is the impact of expert trajectories and the definition of specific predicates for ILP? Some study is needed about this, e.g., considering fewer / more expert trajectories, or with different quality and reliability, or omitting some predicates
4. I do not get figure 6a: in the main text, you write that your agent never violates the pothole constraint; however, I see constraint violation for that only, in the figure.
5. In the context of constrained / safe RL and autonomous driving, typically temporal formulae are used [2]. How does your method compare to this or similar works, where also LTL formulae are learned [3]?

[1] Stooke, Adam, Joshua Achiam, and Pieter Abbeel. "Responsive safety in reinforcement learning by pid lagrangian methods." International Conference on Machine Learning. PMLR, 2020.
[2] Yang, Chaeeun, Sojeong Yoon, and Kyunghoon Cho. "End-to-End Path Planning under Linear Temporal Logic Specifications." IEEE Access (2024).
[3] De Giacomo, Giuseppe, et al. "Imitation learning over heterogeneous agents with restraining bolts." Proceedings of the international conference on automated planning and scheduling. Vol. 30. 2020.

**Anonymity:**

Remain anonymous

---

### Official Review · Reviewer_UarJ · 2025-04-05
**Neuro-Symbolic Inverse Constrained Reinforcement Learning**

**Rating:** 6
**Confidence:** 4

**Review:**

This paper presents a novel integration of a neurosymbolic approach with inverse constrained reinforcement learning (ICRL) to learn interpretable and transferable safety constraints. The key contributions are:
1. Constraint Inference via ICRL: Constraints are inferred by identifying state-action pairs that are likely under a learned policy but rarely seen in expert demonstrations—similar in spirit to Scobee and Sastry (2019). The reward function is used to encode preferences for adhering to these constraints.
2. Learning General, Symbolic Safety Constraints: The learned rules generalize beyond individual state-action pairs, supporting transfer across different environments—an improvement over prior work focused on lower-level constraints.
3. Defeasible Constraints: The paper introduces the concept of defeasible constraints, enabling the system to adapt to new information without retraining from scratch, increasing practical applicability in dynamic settings.

The method is evaluated on a traffic simulation environment where the agent should not go off-road and not move to a junction if the light is red. They further allow violating the first constraint in a subsequent task.

Strengths
1. Leaning Defeasible Constraints seems novel in the literature
2. Learning rules is more interpretable + transferrable to different environment configurations
3. Writing is generally clear, although more details about the implementation would be appreciated
4. Experiments compare against relevant baselines and evaluate both learning hard and defeasible constraints

Weaknesses
- Some missing relevant informationL
    1. It is not clear now the reward function defined - fairly important component
    2. hyperparameters for policy learning

- The motivation behind inferring constraints is unclear: usually (in safe RL), one would want to give the constraints and integrate them into the learning procedure

- A more formal definition of the Constrained Markov Decision Process (CMDP) would improve clarity, especially in Section 2.2.

- The paper omits a discussion of Neural DNF-MT: A Neuro-symbolic Approach for Learning Interpretable and Editable Policies (Baugh et al., 2025). While this does not directly impact the contributions, including it would help position the work more thoroughly in the literature.

- The authors should discuss why another learning methods such as ILASP (Law et al. 2015) are not used. This would  be particularly relevant here given its capability to handle noisy examples. The default assumption might be good strong: just because an action is unlikely in expert demonstrations does not mean it is impossible (just not present in the data). Handling noise could help with this problem more naturally,

**Anonymity:**

Remain anonymous

---

### Official Review · Reviewer_mDAT · 2025-04-05
**NSICRL represents a novel and robust approach. I believe it to be interesting and relevant for the community.**

**Rating:** 7
**Confidence:** 3

**Review:**

This paper presents Neuro-Symbolic Inverse Constrained Reinforcement Learning (NSICRL), a novel framework that integrates Inductive Logic Programming (ILP) with Deep Q-Learning (DQL). The authors aim to simultaneously learn a policy ($\pi$) and defeasible constraints (social norms). Their proposed method leverages expert data to iteratively refine norms by applying penalties to the reward function whenever constraints are violated, but also needs a pre-existing knowledge base

The authors validate their approach in a discrete gridworld scenario using the SUMO traffic simulator. Results demonstrate that the approach successfully integrates deep reinforcement learning and symbolic methods, effectively enforcing constraints while allowing adaptability to new or updated norms.

Pros:
- Innovative and robust integration of neural (DQL) and symbolic (ILP) learning methods.
- Effective handling of norm violations through dynamic penalty adjustments.
- Demonstrated adaptability and interpretability of constraints.

Cons:

- Evaluations restricted to discrete gridworld environments; performance in continuous or stochastic domains remains untested.
- Algorithm effectiveness heavily depends on the quality of expert demonstrations and pre-existing knowledge.
- The complexity of the integrated neuro-symbolic system raises concerns about scalability.

Overall, the paper is clearly written and presents a compelling and novel approach. I recommend acceptance.

**Anonymity:**

Remain anonymous